# Effects of X-ray–based diagnosis and explanation of knee osteoarthritis on patient beliefs about osteoarthritis management: A randomised clinical trial

Belinda J. Lawford[1]*, Kim L. Bennell[1], Dan Ewald[2], Peixuan Li[3,4], Anurika De Silva[3,4], Jesse Pardo[1], Barbara Capewell[5], Michelle Hall[6], Travis Haber[1], Thorlene Egerton[1], Stephanie Filbay[1], Fiona Dobson[1], Rana S. Hinman[1]

1 Centre for Health, Exercise and Sports Medicine, Department of Physiotherapy, School of Health Sciences, The University of Melbourne, Victoria, Australia, 2 University Centre for Rural Health, University of Sydney School of Medicine, Camperdown, Australia, 3 Methods and Implementation Support for Clinical and Health (MISCH) research Hub, Faculty of Medicine, Dentistry and Health Sciences, The University of Melbourne, Melbourne, Australia, 4 Centre for Epidemiology and Biostatistics, Melbourne School of Population and Global Health, The University of Melbourne, Melbourne, Australia, 5 Melbourne, Australia, 6 Sydney Musculoskeletal Health, The Kolling Institute, School of Health Sciences, University of Sydney, New South Wales, Australia

* belinda.lawford@unimelb.edu.au

## Abstract

### Background

Although X-rays are not recommended for routine diagnosis of osteoarthritis (OA), clinicians and patients often use or expect X-rays. We evaluated whether: (i) a radiographic diagnosis and explanation of knee OA influences patient beliefs about management, compared to a clinical diagnosis and explanation that does not involve X-rays; and (ii) showing the patient their X-ray images when explaining radiographic report findings influences beliefs, compared to not showing X-ray images.

### Methods and findings

This was a 3-arm randomised controlled trial conducted between May 23, 2024 and May 28, 2024 as a single exposure (no follow-up) online survey. A total of 617 people aged ≥45 years, with and without chronic knee pain, were recruited from the Australian-wide community. Participants were presented with a hypothetical scenario where their knee was painful for 6 months and they had made an appointment with a general practitioner (primary care physician). Participants were randomly allocated to one of 3 groups where they watched a 2-min video of the general practitioner providing them with either: (i) clinical explanation of knee OA (no X-rays); (ii) radiographic explanation (not showing X-ray images); or (iii) radiographic explanation (showing X-ray images). Primary comparisons were: (i) clinical explanation (no X-rays) versus radiographic explanation (showing X-ray images); and (ii) radiographic explanation (not showing X-ray images) versus radiographic explanation (showing X-ray images). Primary outcomes were perceived (i) necessity of joint

**Data Availability Statement:** Data are available on figshare (https://doi.org/10.6084/m9.figshare.28216649).

**Funding:** This study was supported by the National Health & Medical Research Council (Investigator Grant #2025733 to RSH). KLB, MH, and SF are supported by National Health & Medical Research Council Investigator grants (#1174431, #1172928, and #1194428, respectively). BJL is supported by an Arthritis Australia Fellowship (number not assigned). The funders had no role in study design, data collection and analysis, decision to publish, or preparation of the manuscript.

**Competing interests:** The authors have declared that no competing interests exist.

**Abbreviations:** GP, general practitioner; NRS, numeric rating scale; OA, osteoarthritis; RCT, randomised controlled trial; SD, standard deviation.

replacement surgery; and (ii) helpfulness of exercise and physical activity, both measured on 11-point numeric rating scales (NRS) ranging 0 to 10.

Compared to clinical explanation (no X-rays), those who received radiographic explanation (showing X-ray images) believed surgery was more necessary (mean 3.3 [standard deviation: 2.7] versus 4.5 [2.7], respectively; mean difference 1.1 [Bonferroni-adjusted 95% confidence interval: 0.5, 1.8]), but there were no differences in beliefs about the helpfulness of exercise and physical activity (mean 7.9 [standard deviation: 1.9] versus 7.5 [2.2], respectively; mean difference −0.4 [Bonferroni-adjusted 95% confidence interval: −0.9, 0.1]). There were no differences in beliefs between radiographic explanation with and without showing X-ray images (for beliefs about necessity of surgery: mean 4.5 [standard deviation: 2.7] versus 3.9 [2.6], respectively; mean difference 0.5 [Bonferroni-adjusted 95% confidence interval: −0.1, 1.2]; for beliefs about helpfulness of exercise and physical activity: mean 7.5 [standard deviation: 2.2] versus 7.7 [2.0], respectively; mean difference −0.2 [Bonferroni-adjusted 95% confidence interval: −0.7, 0.3]). Limitations of our study included the fact that participants were responding to a hypothetical scenario, and so findings may not necessarily translate to real-world clinical situations, and that it is unclear whether effects would impact subsequent OA management behaviours.

## Conclusions

An X-ray–based diagnosis and explanation of knee OA may have potentially undesirable effects on people's beliefs about management.

## Trial registration

ACTRN12624000622505.

## Author summary

### Why was this study done?

- Uptake of exercise and physical activity among people with osteoarthritis (OA) is low, and use of joint replacement surgery—recommended only for the minority of people with end-stage disease or whose symptoms cannot be managed adequately nonsurgically—is inappropriately high.

- Although radiographic imaging is not recommended for routine diagnosis of OA, many primary healthcare providers globally still rely on X-rays to diagnose OA.

- We aimed to provide, to our knowledge, the first empirical evidence from a randomised controlled trial to determine if use of X-rays to diagnose and explain OA influences people's beliefs about OA management.

## What did the researchers do and find?

- This was a 3-arm randomised controlled trial involving 617 people who were presented with a hypothetical scenario (that their knee was painful and they had made an appointment with a general practitioner) before being randomly allocated to one of 3 groups where the general practitioner provides them with either: (i) clinical explanation of knee OA (no X-rays); (ii) radiographic explanation (not showing X-ray images); or (iii) radiographic explanation (showing X-ray images).

- We found that using X-rays increased participant beliefs that joint replacement surgery was necessary, but made no difference to beliefs about the helpfulness of exercise and physical activity.

- We also found that there were no differences in beliefs when explaining radiographic report findings with and without showing the X-ray images.

## What do these findings mean?

- Our work provides, to our knowledge, the first causal evidence that using X-rays rather than a clinical approach to diagnose and explain knee OA can have potentially undesirable impacts on patient beliefs about knee OA.

- Changing clinical practice around use of X-rays will require a substantial shift in long-held beliefs and habits and our findings are a first step by showing use of X-rays to diagnose knee OA causally impacts patient beliefs.

- Although many outcomes met or exceeded our a priori minimal 1.0-unit difference immediately after watching the videos, it is unclear whether effects on patient beliefs would impact subsequent OA management behaviours.

## Introduction

Knee osteoarthritis (OA) is a leading cause of pain and physical disability worldwide [1,2]. As there is no cure, engagement in long-term self-management of symptoms is advocated, with exercise and physical activity recommended as core first-line approaches for all people with knee OA [3–6]. However, uptake of exercise and physical activity among people with OA is low, and use of joint replacement surgery—recommended only for the minority of people with end-stage disease or whose symptoms cannot be managed adequately nonsurgically—is inappropriately high [7,8]. Over 1.2 million joint replacements are performed annually in the United States, incurring an estimated $USD20 billion in healthcare costs [9,10]. As joint replacement surgery rates are projected to increase [11,12], strategies to reduce overuse of surgery are needed.

One potential driver of low uptake of nonsurgical treatments, and overuse of joint replacement surgery, is inaccurate public beliefs about OA [13]. As OA is characterised by pain and physical dysfunction [6,14], people with OA often assume their symptoms stem from joint structural changes [15]. However, research shows that joint structural changes are poorly

correlated with pain and physical dysfunction, and do not necessarily predict prognosis of symptoms [14,16,17].

Accordingly, radiographic imaging is not recommended for routine diagnosis of OA [14,16,18], and instead, a clinical diagnosis is advocated based on age and symptom presentation [6]. Despite this, many primary healthcare providers globally still rely on X-rays to diagnose OA [8,19,20] and people with musculoskeletal conditions, including OA, expect imaging as part of their clinical care [21–23]. Qualitative studies suggest that, when people with OA are shown X-rays, they believe they need to protect their joint from further damage [21,24], and therefore, may be less willing to engage in first-line nonsurgical care like exercise. However, no empirical evidence from randomised controlled trials (RCTs) exists to determine if use of X-rays to diagnose and explain OA influences people's beliefs about OA management. Our primary aims were to evaluate whether: (i) a radiographic diagnosis and explanation of knee OA influences patient beliefs about OA management, compared to a clinical diagnosis and explanation that does not involve X-rays; and (ii) showing the patient X-ray images when explaining radiographic report findings influences beliefs, compared to not showing X-rays.

## Methods

### Study design and participants

This was an online 3-arm RCT using a single exposure (no follow-up). Given the practical challenges of identifying and recruiting people when they first present to care seeking a diagnosis for chronic knee pain, we used a hypothetical scenario approach. The trial was prospectively registered (Australian New Zealand Clinical Trials Registry #12624000622505 https://www.anzctr.org.au/Trial/Registration/TrialReview.aspx?id=387341&isReview=true) and the protocol can be found in S1 Appendix. The Institutional Human Research Ethics Committee approved the study and participants provided informed consent. This study is reported as per CONSORT guidelines [25] (S2 Appendix).

Participants were recruited from across Australia (in any State or Territory) via a consumer network of people who self-nominated to participate in digital survey-based research (Cint Pty Ltd). Participants receive a small financial incentive (approximately $2.55 AUD for completion of a 15- to 20-min survey) from Cint to compensate them for their time and participation. Incentives were only provided after completing the entire survey to the end. Participants were eligible if they: (i) lived in Australia; (ii) were aged ≥45 years (to reflect clinical criteria for OA [6]); (iii) had never consulted a clinician for chronic knee pain; and (iv) were able to read and understand English. Based on similar studies [26–29], we recruited an equal number of people who had, or had not, experienced activity-related knee pain over the prior 3 months. People with and without knee pain were included to simulate the situation of someone without a prior diagnosis of knee OA, and with minimal preconceived knowledge about OA and its treatment, presenting to their general practitioner for the first time. This also allowed us to conduct an exploratory analysis to determine if lived experience of knee pain had a treatment-moderating effect on primary outcomes.

### Randomisation and masking

Participants were randomised (1:1:1 ratio, stratified by whether or not they had knee pain) to one of 3 groups using the randomiser function in Qualtrics set to "evenly present elements." Randomisation order was unknown to researchers. Although participants (who were the assessors as outcomes were self-reported) were unblinded to their allocated intervention, limited disclosure blinded them to the alternate intervention groups. Participants were informed that the trial compared different ways of diagnosing OA, but we did not disclose specific

intervention components or study hypotheses. The statistical analysis plan was completed while biostatisticians were blinded to group allocation and published online (https://healthsciences.unimelb.edu.au/departments/physiotherapy/chesm/research-overview/knee-osteoarthritis-diagnosis-trial) and in S3 Appendix.

## Procedures

The trial was delivered via online survey software (Qualtrics, Utah, United States of America). After confirming eligibility, participants completed demographic questions (S4 Appendix). They were then asked to consider a hypothetical scenario where their knee had been painful for 6 months and they had made an appointment with a general practitioner (GP; primary care physician) to determine what was wrong (S5 Appendix). Participants were then randomised to one of 3 groups where they watched a ~2-min video of a GP (author DE, who featured in all 3 videos) providing them with a hypothetical: (i) clinical explanation (no X-rays); (ii) radiographic explanation (not showing X-ray images); or (iii) radiographic explanation (showing X-ray images). The hypothetical scenario and all video scripts (Table 1) were created by the researchers (including BC, a consumer with lived experience of knee OA, and DE, a practicing GP) to ensure they reflected typical clinical encounters. Videos were filmed by the researchers in DE's consulting room. Participants were required to confirm (via checkbox) that they had watched the whole video before completing outcome measures. Videos can be found in S1, S2 and S3 Videos.

**Table 1. Video scripts for each of the 3 groups.**

| Group | Clinical explanation (no X-rays) | Radiographic explanation (not showing X-ray images) | Radiographic explanation (showing X-ray images) |
|---|---|---|---|
| Diagnosis and explanation | The most common cause of knee pain in people aged 45 and over is osteoarthritis. Based on your age and the nature of your symptoms, such as having pain with activity like you have described, I think osteoarthritis is the cause of your knee pain. Special tests—such as an X-ray—are not recommended unless we need to rule out another cause of your knee pain. We would only do that if you had unusual symptoms—which you do not have. Also, X-rays do not tell us what will happen with your knee in the future, and we can recommend treatments for you without needing an X-ray. X-rays also expose you to radiation, and it is best to minimise your exposure. | "It sounds like you could have knee osteoarthritis—I'm going to send you for X-rays of your knee to confirm what is going on in your joint." . . . "I have the report from the X-rays of your knee, which show you do have osteoarthritis, like I suspected. The report states that, in your painful knee, there is reduction in the joint space—which means that the area of the knee joint between the bones is narrower, showing that the cartilage is thinner. The report also states that there are osteophytes—or bony spurs—that are growing around the edges of the joint. | "It sounds like you could have knee osteoarthritis—I'm going to send you for X-rays of your knee to confirm what is going on in your joint." . . . "Here are the X-rays of your knee (Image A*), which show you do have osteoarthritis, like I suspected. As you can see here (Image B*), this is your normal knee—you can see the nice space between the bones in your joint indicating healthy cartilage. This (Image C*) is your painful knee, which shows you have osteoarthritis. This area of the knee joint between the bones is narrower, showing where the cartilage is thinner. Over here (Image D*), you can see some osteophytes—or bony spurs—that are growing around the edges of the joint. |
| Information about prognosis and treatment (provided to ALL groups) | Osteoarthritis is a condition of the whole knee, including surrounding cartilage, bones, ligaments, and muscles. Initially, changes occur to some of the structures in the knee. This may be due to previous injuries, genetics, or the type of work you have done. The body will then try to repair these changes. While these repair processes can lead to further joint changes, they often keep the knee working normally. Some people have lots of changes in their knee joint, but this does not necessarily mean they have more knee pain. For most people, osteoarthritis will not get progressively worse. In fact, only about a third of people will get worse over time. There are treatments that can provide pain relief and allow you to get on with your life. Usually, treatment includes an exercise programme tailored to your condition and ability, medicines like paracetamol or nonsteroidal anti-inflammatory drugs, and weight loss if necessary. Some people may need joint replacement surgery if those treatments are not effective. I am sure you have many questions about your knee pain after hearing all that. I am happy to answer any questions you have. | | |

* X-ray images shown in S10 Appendix.

Videos shown in S1, S2 and S3 Videos.

### Clinical explanation (no X-rays)

The GP provided a clinical diagnosis and explanation of OA based on age ($\geq$45 years) and nature of symptoms (activity-related joint pain and either no morning stiffness or morning stiffness that lasts no longer than 30 min), as recommended by the National Institute for Health and Care Excellence guidelines [6]. The GP then provided an evidence-based explanation of what OA is, its prognosis, and how it can be managed (adapted from Arthritis Australia website information [30]). The video was 1 min and 46 s in length.

### Radiographic explanation (not showing X-ray images)

The GP told the participant that they might have OA and that an x-ray was needed for confirmation. Participants were asked to imagine that they had gone to get an X-ray before returning to the GP for the results. The GP provided a summary explanation of the X-ray report (without showing the participant the X-ray images), including non-emotive, objective lay descriptions about the X-ray findings of narrowed joint space and osteophytes. The GP then explained what OA is, its prognosis, and how it can be managed using the exact same script as in the clinical explanation (no X-rays) group. The video was 1 min and 52 s in length.

### Radiographic explanation (showing X-ray images)

The video in this group was almost identical to the radiographic explanation (not showing X-ray images) group, except that participants were shown the X-ray images while the GP provided the explanation of the findings. The GP then provided an explanation about what OA is and how it can be managed using the exact same script as for the other 2 groups. This video was 1 min and 58 s in length.

### Outcomes

Bespoke outcome measures (S6 Appendix) were adapted from our similar online RCTs evaluating information effects on patient beliefs [28,29,31]. Primary outcomes were belief about the necessity of joint replacement surgery ("Based on the video you have just watched, do you think joint replacement surgery (to replace the affected joint with an artificial joint) would be necessary for your hypothetical knee osteoarthritis at some stage?") and helpfulness of exercise and physical activity ("Based on the video you have just watched, do you think exercise and physical activity would be helpful to manage your hypothetical knee osteoarthritis?"), each rated on an 11-point numeric rating scale (NRS) ranging from 0 = "definitely not necessary/helpful" to 10 = "definitely necessary/helpful."

Secondary outcomes (S6 Appendix) included beliefs about the helpfulness of medication in managing knee OA, beliefs that exercise and physical activity would damage the knee, level of concern that their hypothetical knee OA would get worse in the future, and fear of movement (Brief Fear of Movement Scale [32]). We evaluated perceptions about the helpfulness of different healthcare providers, including orthopaedic surgeons, rheumatologists, and physiotherapists. Finally, participants were asked how satisfied they were with the hypothetical GP consultation and the information provided and their confidence in the accuracy of the diagnosis that they received.

### Process measures

We collected data from Qualtrics that measured the time (in seconds) that participants spent on the survey page with the video. Participants indicated (yes/no) whether the video audio and visual quality was sufficient for them to watch and understand.

## Sample size

A sample of 609 (203 per group) was required to detect a minimum difference of 1.0 NRS units between groups in each primary outcome with 80% power and a two-sided alpha of 0.0125 (conservative Bonferroni correction for 2 primary outcomes and 2 primary pair-wise comparisons), assuming a between-participant standard deviation (SD) of 3.0 [28] and no attrition [31]. As there is no established minimal important difference for either primary outcome, we chose to detect a minimum difference of 1.0 unit, which corresponds to 10% of the maximum score and was used in similar trials in OA and low back pain [28,29,31,33].

## Statistical analysis

Our primary comparisons were: (i) clinical explanation (no X-rays) versus radiographic explanation (showing X-ray images); and (2) radiographic explanation (not showing X-ray images) versus radiographic explanation (showing X-ray images). Our secondary comparison was clinical explanation (no X-rays) versus radiographic explanation (not showing X-ray images).

Data were analysed in Stata version 17.0 [34]. As <5% of data for each primary outcome was missing, multiple imputation was not applied, and analyses were performed on complete case data. Separate linear regression models were used to compare post-intervention scores between groups for each primary and secondary outcome. Regression assumptions, including those of linearity and homoscedasticity, were assessed using standard diagnostic plots. All analysis models were adjusted for the stratification factor lived experience of knee pain.

We interpreted each outcome by examining the estimated between-group mean differences, using two-sided 95% CIs and $p$-values (Bonferroni-adjusted for primary outcomes across primary comparisons), reported in accordance with recommendations of the American Statistical Association [35].

To explore whether lived experience of knee pain moderated intervention effects, separate linear regression models for each primary outcome were fitted with terms included for group, presence/absence of recent knee pain, and an interaction between the two.

## Results

From the 1,372 people screened for eligibility, we enrolled 617 participants between May 23 and May 28, 2024 (Fig 1). Major reasons for exclusion were previously having consulted a clinician for knee pain ($n$ = 523, 36%), discontinuing the survey before randomisation ($n$ = 152, 20%), or not consenting to participate ($n$ = 37, 5%). Characteristics of participants were similar between groups (Table 2). There were no differences between participants who did and did not complete primary outcomes (S7 Appendix).

### Primary pair-wise comparisons: Primary outcomes

Compared to clinical explanation (no X-rays), those who received radiographic explanation (showing X-ray images) believed joint replacement surgery was more necessary (mean difference 1.1 NRS units [Bonferroni-adjusted 95% CI: 0.5, 1.8]; $p$ < 0.001; Table 3), which exceeded our a priori minimum difference of 1-unit. This outcome did not differ between radiographic explanation with and without showing X-ray images. We did not observe any pair-wise differences in beliefs about the helpfulness of exercise and physical activity.

### Primary pair-wise comparisons: Secondary outcomes

Compared to clinical explanation (no X-rays), those who received radiographic explanation (showing X-ray images) believed exercise and physical activity was more damaging (0.6 [95%

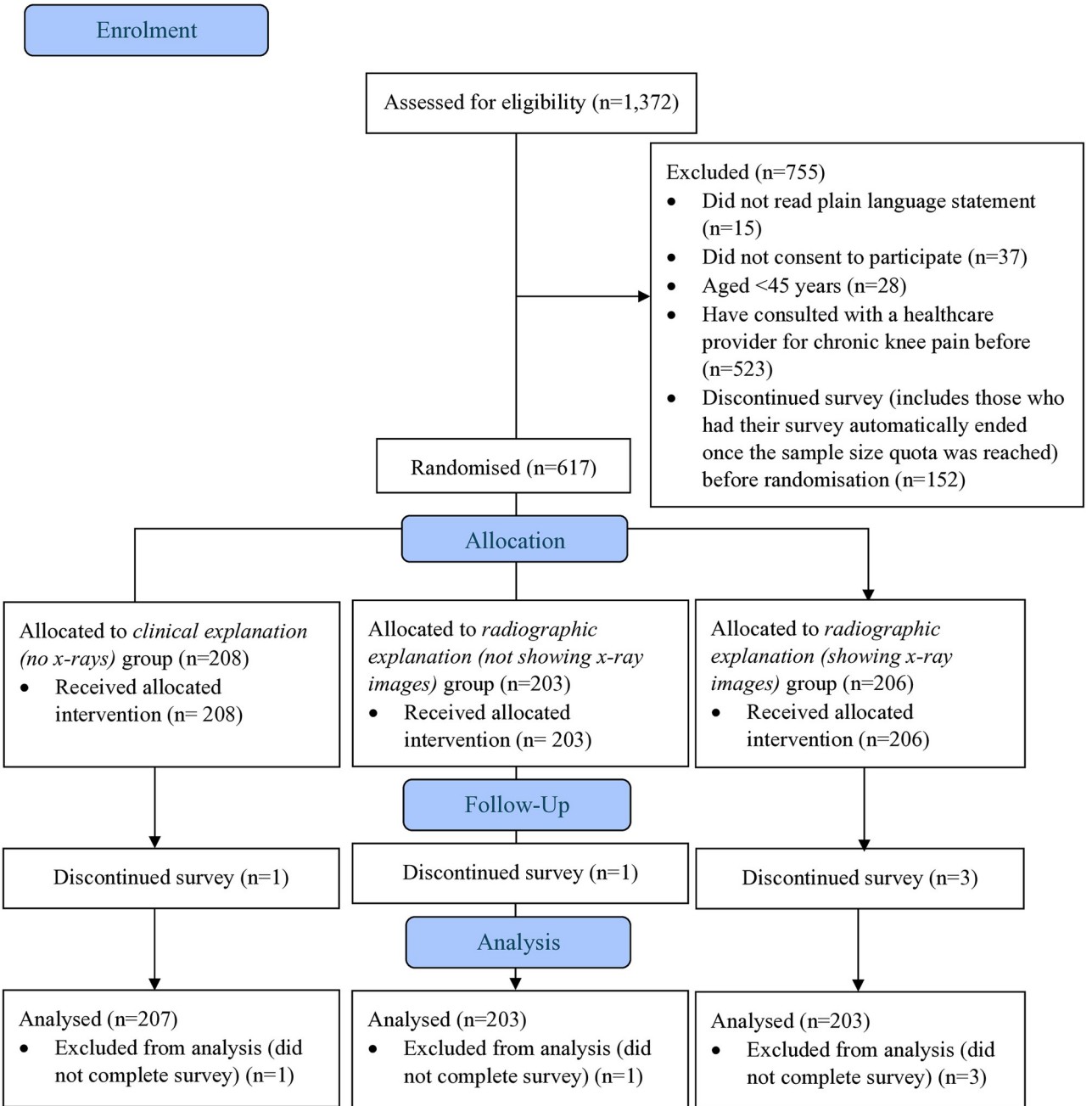

**Fig 1. Participant flow through the trial.**

CI: 0.2, 1.1]; $p$ = 0.008; Table 3), were more concerned about their knee problem getting worse (1.0 [0.5, 1.5]; $p$ = $p$ < 0.001), and had greater fear of movement (1.0 [0.3, 1.7]; $p$ = 0.005). Participants who received radiographic explanation (showing X-ray images) also believed an orthopaedic surgeon (1.3 [0.8, 1.8]; $p$ < 0.001) rheumatologist (0.8 [0.2, 1.3]; $p$ = 0.004) and physiotherapist (0.6 [0.1, 1.0]; $p$ = 0.009) would be more helpful, were more satisfied with the consultation (0.4 [0.1, 0.8]; $p$ = 0.023), were more satisfied with the information provided by the GP (0.4 [0.0, 0.7]; $p$ = 0.033), and were more confident in the accuracy of their diagnosis

**Table 2. Baseline characteristics of participants by group, reported as mean (standard deviation) unless otherwise stated.**

| | Clinical explanation (no X-rays) | Radiographic explanation (not showing X-ray images) | Radiographic explanation (showing X-ray images) |
|---|---|---|---|
| | [*N* = 208] | [*N* = 203] | [*N* = 206] |
| Age (years), median (IQR) | 60.0 (50.0–67.0) | 61.0 (51.0–69.0) | 59.0 (50.0–68.0) |
| Gender, *N* (%) | | | |
| Male | 89 (43) | 89 (44) | 97 (47) |
| Female | 118 (57) | 114 (56) | 109 (53) |
| Transgender male | 1 (<1) | 0 (0) | 0 (0) |
| Ethnicity, *N* (%) | | | |
| Australian/New Zealand | 148 (71) | 137 (67) | 145 (70) |
| Aboriginal and Torres Strait Islander | 5 (2) | 2 (1) | 2 (1) |
| European | 36 (17) | 40 (20) | 35 (17) |
| Asian | 9 (4) | 14 (7) | 14 (7) |
| Other Oceania | 1 (1) | 0 (0) | 1 (1) |
| North African and Middle Eastern | 2 (1) | 2 (1) | 2 (1) |
| Sub-Saharan Africa | 0 (0) | 0 (0) | 2 (1) |
| North American | 2 (1) | 3 (1) | 1 (<1) |
| South American | 1 (<1) | 2 (1) | 0 (0) |
| Other | 3 (1) | 3 (1) | 3 (1) |
| Prefer not to answer | 1 (<1) | 0 (0) | 1 (<1) |
| State/territory, *N* (%) | | | |
| Australian Capital Territory | 3 (1) | 2 (1) | 3 (1) |
| New South Wales | 57 (27) | 52 (26) | 63 (31) |
| Northern Territory | 2 (1) | 1 (<1) | 1 (<1) |
| Queensland | 52 (25) | 43 (21) | 44 (21) |
| South Australia | 21 (10) | 19 (9) | 14 (7) |
| Tasmania | 5 (2) | 4 (2) | 6 (3) |
| Victoria | 52 (25) | 69 (34) | 53 (26) |
| Western Australia | 16 (8) | 13 (6) | 22 (11) |
| Height (m) | 1.7 (0.1) | 1.7 (0.1) | 1.7 (0.1) |
| Weight (kg) | 78.8 (19.1) | 78.0 (17.7) | 80.7 (20.3) |
| Body mass index (kg/m$^2$) | 27.8 (6.5) | 28.1 (7.8) | 28.7 (7.5) |
| Highest education level, *N* (%) | | | |
| Primary school | 3 (1) | 3 (1) | 3 (1) |
| Secondary school | 76 (37) | 63 (31) | 65 (32) |
| Trade or trade certificate | 49 (24) | 49 (24) | 48 (23) |
| University or tertiary institute degree | 65 (31) | 65 (32) | 75 (36) |
| Higher university degree (e.g. Masters, PhD) | 15 (7) | 23 (11) | 14 (7) |
| Do not know/unsure | 0 (0) | 0 (0) | 1 (<1) |
| Financial situation, *N* (%) | | | |
| Find it a strain to get by from week to week | 44 (21) | 22 (11) | 21 (10) |
| Have to be careful with money | 81 (39) | 87 (43) | 83 (40) |
| Able to manage without much difficulty | 49 (24) | 59 (29) | 52 (25) |
| Quite comfortably off | 28 (13) | 24 (12) | 34 (17) |
| Very comfortably off | 6 (3) | 9 (4) | 14 (7) |
| Prefer not to answer | 0 (0) | 2 (1) | 2 (1) |
| Participation in regular exercise/physical activity, *N* (%) | | | |
| None | 57 (27) | 65 (32) | 56 (27) |

*(Continued)*

**Table 2.** (Continued)

| | Clinical explanation (no X-rays) | Radiographic explanation (not showing X-ray images) | Radiographic explanation (showing X-ray images) |
|---|---|---|---|
| | [*N* = 208] | [*N* = 203] | [*N* = 206] |
| Yes, 1 time per week | 21 (10) | 23 (11) | 23 (11) |
| Yes, 2–3 times per week | 52 (25) | 63 (31) | 63 (31) |
| Yes, 4–5 times per week | 52 (25) | 35 (17) | 37 (18) |
| Yes, 6+ times per week | 26 (12) | 17 (8) | 27 (13) |
| Activity-related knee pain in the last 3 months, *N* (%) | 104 (50) | 101 (50) | 102 (50) |
| Joint with knee pain*, *n* (%) | | | |
| Left knee only | 30 (29) | 31 (31) | 29 (28) |
| Right knee only | 35 (34) | 32 (32) | 36 (35) |
| Both knees | 39 (38) | 38 (38) | 37 (36) |
| Self-reported knee pain (NRS) | 4.5 (2.0) | 4.3 (2.3) | 4.6 (2.2) |
| Self-reported physical function (NRS) | 3.9 (2.5) | 3.9 (2.7) | 4.3 (2.5) |
| Regular pain relief for musculoskeletal condition, *N* (%) | 62 (30) | 55 (27) | 68 (33) |
| Ability to read and understand written health information#, median (IQR) | 4.0 (4.0–5.0) | 4.0 (4.0–5.0) | 4.0 (4.0–5.0) |

IQR, interquartile range (25th to 75th percentile); kg, kilograms; m, metres; NRS, numerical rating scale ranging from 0 ("no pain" or "no interference") to 10 ("worst pain possible" or "maximal interference with function").

* The denominator for joint with knee pain was the number of participants with knee pain in each group.

# Rated using a 5-point scale with terminal descriptors of 1 = "always difficult" to 5 = "always easy."

(0.6 [0.3, 1.0]; *p* = 0.001) than those who received clinical explanation (no X-rays). There were no differences in any secondary outcome when comparing radiographic explanation with and without showing X-ray images.

## Secondary pair-wise comparisons

Compared to clinical explanation (no X-rays), those who received radiographic explanation (not showing X-ray images) believed joint replacement surgery was more necessary (0.6 [0.1, 1.1]; *p* = 0.020; Table 3), were more concerned about their knee problem getting worse (0.7 [0.2, 1.2]; *p* = 0.003), believed an orthopaedic surgeon (1.3 [0.8, 1.8]; *p* < 0.001) and rheumatologist (0.7 [0.2, 1.2]; *p* = 0.007) would be more helpful, and were more confident in the accuracy of the diagnosis (0.5 [0.1, 0.9]; *p* = 0.011).

## Subgroup analyses

There were no differences in outcomes between participants with and without recent lived experience of knee pain (S8 Appendix).

## Process measures

Most (99%) participants believed the video and audio quality was sufficient (S9 Appendix). Participants spent a median (interquartile range) of 2.2 (2.1 to 2.4) min watching the videos.

## Discussion

We evaluated the effects of a radiographic knee OA diagnosis and explanation delivered by a GP on participant beliefs about osteoarthritis management, compared to a clinical explanation

**Table 3. Summary measures and estimated between-group mean differences [95% CI] for primary and secondary outcomes using complete case data.**

| | Post-intervention, mean (SD) | | | Between-group difference in post-intervention scores | | | | | |
| --- | --- | --- | --- | --- | --- | --- | --- | --- | --- |
| | Group 1 | Group 2 | Group 3 | Primary comparison 1 Group 3 vs. 1 | | Primary comparison 2 Group 3 vs. 2 | | Secondary comparison Group 2 vs. 1 | |
| | Clinical explanation (no X-rays) [N = 208][a] | Radiographic explanation (not showing X-ray images) [N = 203][a] | Radiographic explanation (showing X-ray images) [N = 206][a] | Estimate[b] (95% CI)[c] | P-value[c] | Estimate[b] (95% CI)[c] | P-value[c] | Estimate[b] (95% CI) | P-value |
| **Primary outcomes** | | | | | | | | | |
| Belief joint replacement surgery is necessary at some stage* | 3.3 (2.7) | 3.9 (2.6) | 4.5 (2.7) | 1.1 (0.5, 1.8) | <0.001 | 0.5 (−0.1, 1.2) | 0.206 | 0.6 (0.1, 1.1) | 0.020 |
| Belief exercise and physical activity is helpful¥ | 7.9 (1.9) | 7.7 (2.0) | 7.5 (2.2) | −0.4 (−0.9, 0.1) | 0.116 | −0.2 (−0.7, 0.3) | 1.370 | −0.2 (−0.6, 0.1) | 0.205 |
| **Secondary outcomes** | | | | | | | | | |
| Belief exercise would damage knee^β | 3.7 (2.5) | 3.9 (2.5) | 4.4 (2.4) | 0.6 (0.2, 1.1) | 0.008 | 0.4 (−0.1, 0.9) | 0.083 | 0.2 (−0.3, 0.7) | 0.355 |
| Belief medication is helpful¥ | 7.6 (2.0) | 7.4 (1.9) | 7.3 (2.0) | −0.2 (−0.6, 0.2) | 0.253 | −0.1 (−0.5, 0.3) | 0.622 | −0.1 (−0.5, 0.3) | 0.512 |
| Level of concern§ | 4.8 (2.5) | 5.5 (2.4) | 5.8 (2.5) | 1.0 (0.5, 1.5) | <0.001 | 0.3 (−0.2, 0.8) | 0.232 | 0.7 (0.2, 1.2) | 0.003 |
| Fear of movement£ | 13.0 (3.4) | 13.6 (3.3) | 14.0 (3.7) | 1.0 (0.3, 1.7) | 0.005 | 0.5 (−0.2, 1.1) | 0.187 | 0.6 (−0.1, 1.2) | 0.095 |
| Belief about helpfulness of orthopaedic surgeon√ | 5.1 (2.9) | 6.4 (2.4) | 6.4 (2.4) | 1.3 (0.8, 1.8) | <0.001 | 0.0 (−0.5, 0.4) | 0.928 | 1.3 (0.8, 1.8) | <0.001 |
| Belief about helpfulness of rheumatologist√ | 5.3 (2.9) | 6.0 (2.3) | 6.0 (2.4) | 0.8 (0.2, 1.3) | 0.004 | 0.1 (−0.4, 0.5) | 0.804 | 0.7 (0.2, 1.2) | 0.007 |
| Belief about helpfulness of physiotherapist√ | 6.6 (2.5) | 7.0 (1.9) | 7.2 (1.9) | 0.6 (0.1, 1.0) | 0.009 | 0.2 (−0.2, 0.5) | 0.373 | 0.4 (0.0, 0.8) | 0.070 |
| Overall satisfaction with consultationα | 7.5 (2.0) | 7.7 (1.9) | 7.9 (1.6) | 0.4 (0.1, 0.8) | 0.023 | 0.2 (−0.1, 0.6) | 0.224 | 0.2 (−0.2, 0.6) | 0.296 |
| Satisfaction with information providedα | 7.6 (2.0) | 7.8 (1.7) | 8.0 (1.5) | 0.4 (0.0, 0.7) | 0.033 | 0.2 (−0.1, 0.5) | 0.193 | 0.2 (−0.2, 0.5) | 0.380 |
| Confidence in accuracy of diagnosisπ | 7.3 (2.1) | 7.8 (1.7) | 8.0 (1.6) | 0.6 (0.3, 1.0) | 0.001 | 0.1 (−0.2, 0.5) | 0.423 | 0.5 (0.1, 0.9) | 0.011 |

CI, confidence interval; SD, standard deviation.

Positive between-group differences indicate higher score in: Group 3 (for Group 3 vs. 1 and Group 3 vs. 2); Group 2 (for Group 2 vs. 1).

[a] Five participants missing all primary and secondary outcomes (3 in Group 1, 1 in Group 2, and 1 in Group 3).

[b] Mean difference (95% CI) in post-intervention scores between groups, adjusted for the stratification factor lived experience with knee pain, estimated using separate linear regression models for each outcome.

[c] For the primary comparisons of the primary outcomes, two-sided 95% CIs and p-values were multiplicity adjusted (alpha = 0.0125).

* Measured using the 11-point numerical rating scale ranging from 0 = "definitely not necessary" to 10 = "definitely necessary."

¥ Measured using the 11-point numerical rating scale ranging from 0 = "definitely not helpful" to 10 = "definitely helpful."

β Measured using the 11-point numerical rating scale ranging from 0 = "definitely would not damage it" to 10 = "definitely would damage it."

§ Measured using the 11-point numerical rating scale ranging from 0 = "not concerned" to 10 = "very concerned."

£ Measured using the Brief Fear of Movement Scale. Scores range 6–24. Higher scores indicate greater fear of movement.

√ Measured using the 11-point numerical rating scale ranging from 0 = "definitely could not help" to 10 = "definitely could help."

α Measured using the 11-point numerical rating scale ranging from 0 = "definitely not satisfied" to 10 = "definitely satisfied."

π Measured using the 11-point numerical rating scale ranging from 0 = "not at all confident" to 10 = "very confident."

that did not use X-rays as the basis of diagnosis. We found that using X-rays increased participant beliefs that joint replacement surgery was necessary, but made no difference to beliefs about the helpfulness of exercise and physical activity. We also found that there were no differences in beliefs when explaining radiographic report findings with and without showing the X-ray images.

Our work provides, to our knowledge, the first causal evidence that using X-rays rather than a clinical approach to diagnose and explain knee OA can have potentially undesirable impacts on patient beliefs about knee OA. Prior qualitative research found that people with OA who receive X-rays of their joint believe that they need to protect it from further damage [21,24,36]. Our work also agrees with prior research showing that people who are given pathoanatomical OA educational information have unfavourable beliefs about OA and its management [15,28,29,33,37]. For participants in our study, receiving an X-ray–based diagnosis and explanation of knee OA may have reinforced existing misconceptions about OA given by the prevailing pathoanatomical discourse in current consumer OA information, which often employs biomedical terminology (e.g., "cartilage degeneration," "bony spurs") and visual representations that depict the joint as "damaged" [30,38–41]. Our findings in knee OA align with research in low back pain. An RCT utilising a hypothetical scenario in low back pain found that a diagnosis and explanation based on magnetic resonance imaging worsened perceptions of back pain severity and fear of movement compared to a diagnosis and explanation not based on imaging [42]. Collectively, these findings may help to increase our understanding of why joint replacement surgery is overused and exercise and physical activity underused in management of knee OA [8].

Although a clinical diagnosis and explanation of knee OA led to potentially more favourable beliefs about OA management, we found that people who received a radiographic diagnosis and explanation were slightly more satisfied with their hypothetical GP consultation and the information provided, and were more confident in the accuracy of the diagnosis. This suggests that people with knee pain expect, and are re-assured, by diagnostic knee radiographs, even when the GP explains that X-ray findings are not necessarily related to knee pain nor predict prognosis. Our recent RCT evaluating different diagnostic labels for persistent hip pain also found that people were more satisfied with a label and explanation that included a description of pathoanatomical changes within the joint [28]. Collectively, these findings may suggest that people with joint pain attribute their pain to structural pathology and desire a biomedical explanation of the cause of their pain [43–45]. Future research should explore ways in which an OA diagnosis and explanation can be conveyed in a manner that supports favourable patient attitudes towards OA, while also satisfying patient expectations for biomedical diagnostic information. Our findings contrast to those in low back pain [42], where participants were more satisfied with a diagnosis that did not rely on imaging. This may be because that study's non-imaging intervention included other elements of best practice care (e.g., information about expectations for recovery, reassurance to reduce concern, and guidance to assist return to usual activity) which were not part of the experimental group that received an imaging-based diagnosis [42].

The magnitude of our between-group differences are similar to those observed in other RCTs utilising hypothetical situations and similar outcomes in OA [28,29,33,46] and other musculoskeletal conditions [26,27,31]. Many of our between-group differences met or exceeded the a priori 1-unit difference we aimed to detect, representing a 10% change in the NRS scale range (which has been suggested to be clinically important for other outcomes on 11-point NRS scales, such as pain intensity [47]). However, it is not clear whether these differences translate to changes in behaviour and improved health outcomes.

Our findings provide empirical evidence of the potential undesirable effects of using X-rays to diagnose and explain OA (S11 Appendix). Clinical guidelines and care standards advocate against routine use of imaging for diagnosing OA because there is no evidence it is beneficial for diagnosis [6,18,48], instead, recommending that healthcare providers diagnose knee OA clinically whenever possible [18]. However, trends in use of imaging for knee OA in Australia, the UK, and US have remained largely unchanged over time [8,19,48]. In the UK, knee radiographs have been ranked as one of the most overused tests in primary care [49]. In the US, X-rays are used for diagnosis of OA in 87% of patients presenting to primary care [19] and in Australia, 46% and 56% of patients presenting to a general practitioner for management of knee or hip OA respectively are referred for imaging [8]. It is thus not surprising that qualitative research in Australia has shown that most general practitioners prefer to use imaging to confirm a knee OA diagnosis, and that they describe OA to their patients as a problem of cartilage degeneration, joint space narrowing on X-ray, or "wear and tear" [22]. When general practitioners do provide a diagnosis of OA, they often avoid the term OA in favour of "wear and tear" [50] or use terms taken directly from radiology reports [51], which can convey a range of negative meanings to patients [52].

General practitioners may decide to order imaging for numerous reasons, including lack of time to discuss the risks and potential harms of imaging versus benefits, concern about malpractice, a desire to meet patient expectations, perceived need to exclude other diagnoses, or lack of confidence to diagnose OA without X-ray [22,53]. Changing clinical practice will require a substantial shift in long-held beliefs and habits [22] and our findings are a first step by showing that use of X-rays to diagnose and explain knee OA causally impacts patient beliefs about OA management. If X-ray is necessary, strategies to reduce potentially undesirable effects on patient beliefs may include reassuring patients that pathoanatomical changes do not necessarily predict prognosis and adapting the language and terminology used in X-ray reports (e.g., including epidemiological information about age-relevant prevalence of imaging characteristics in asymptomatic individuals) [42,54,55].

Our study has limitations. Participants with and without knee pain were asked to respond to a hypothetical scenario. While we found no moderating effect of the lived experience of recent knee pain on outcomes, our findings may not necessarily translate to real-world clinical situations. Although many outcomes met or exceeded our a priori minimal 1.0-unit difference immediately after watching the videos, it is unclear whether these effects on patient beliefs would impact their subsequent OA treatment. As we included only people who spoke and read English and those who had never previously consulted a clinician for chronic knee pain, our findings may not be generalisable to non-English speaking people or those who have sought care for chronic knee pain.

## Conclusion

An X-ray–based diagnosis and explanation of knee osteoarthritis may have potentially undesirable effects on people's beliefs about management.

## Supporting information

**S1 Appendix. Trial protocol.**
(DOCX)

**S2 Appendix. CONSORT checklist.**
(DOCX)

**S3 Appendix. Statistical analysis plan.**
(DOCX)

**S4 Appendix. Baseline descriptive demographic measures.**
(DOCX)

**S5 Appendix. Hypothetical scenario participants were asked to consider before being randomised to a group.**
(DOCX)

**S6 Appendix. Primary and secondary outcomes and process measures.**
(DOCX)

**S7 Appendix. Baseline characteristics of participants who did and did not complete at least one primary outcome, reported as mean (standard deviation) unless otherwise stated.**
(DOCX)

**S8 Appendix. Moderation of intervention effect by lived experience of knee pain on primary outcomes using complete case data.**
(DOCX)

**S9 Appendix. Process measures.**
(DOCX)

**S10 Appendix. X-ray images shown in the radiographic explanation (showing X-rays) group.**
(DOCX)

**S11 Appendix. Summary of key findings and implications for clinical practice.**
(TIF)

**S1 Video. Clinical explanation (no X-rays).**
(MP4)

**S2 Video. Radiographic explanation (not showing X-ray images).**
(MP4)

**S3 Video. Radiographic explanation (showing images).**
(MP4)

## Acknowledgments

Mr. Alexander Kimp for filming and editing the intervention videos.

## Author Contributions

**Conceptualization:** Belinda J. Lawford, Kim L. Bennell, Dan Ewald, Barbara Capewell, Rana S. Hinman.

**Data curation:** Belinda J. Lawford.

**Formal analysis:** Peixuan Li, Anurika De Silva.

**Funding acquisition:** Rana S. Hinman.

**Investigation:** Belinda J. Lawford, Kim L. Bennell, Jesse Pardo, Barbara Capewell, Michelle Hall, Travis Haber, Thorlene Egerton, Stephanie Filbay, Fiona Dobson, Rana S. Hinman.

**Methodology:** Belinda J. Lawford, Kim L. Bennell, Dan Ewald, Jesse Pardo, Barbara Capewell, Michelle Hall, Travis Haber, Thorlene Egerton, Stephanie Filbay, Fiona Dobson, Rana S. Hinman.

**Project administration:** Belinda J. Lawford, Jesse Pardo.

**Resources:** Kim L. Bennell, Dan Ewald, Jesse Pardo.

**Writing – original draft:** Belinda J. Lawford, Rana S. Hinman.

**Writing – review & editing:** Belinda J. Lawford, Kim L. Bennell, Dan Ewald, Peixuan Li, Anurika De Silva, Jesse Pardo, Barbara Capewell, Michelle Hall, Travis Haber, Thorlene Egerton, Stephanie Filbay, Fiona Dobson, Rana S. Hinman.

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
