## [Editor Report · Decision Letter 0]

30 Sep 2024

Dear Dr Lawford, 

Thank you for submitting your manuscript entitled "Effects of x-ray based diagnosis and explanation of knee osteoarthritis on patient beliefs about osteoarthritis management: A randomised clinical trial." for consideration by PLOS Medicine.

Your manuscript has now been evaluated by the PLOS Medicine editorial staff as well as by an academic editor with relevant expertise, and I am writing to let you know that we would like to send your submission out for external peer review.

Please re-submit your manuscript within two working days, i.e. by October 2nd.

Feel free to email me directly at hvanepps@plos.org if you have any queries relating to your submission.

Kind regards,

Heather Van Epps, PhD

Executive Editor

PLOS Medicine

---

## [Decision Letter · Decision Letter 1]

12 Dec 2024

Dear Dr Lawford,

Many thanks for submitting your manuscript "Effects of x-ray based diagnosis and explanation of knee osteoarthritis on patient beliefs about osteoarthritis management: A randomised clinical trial." (PMEDICINE-D-24-03202R1), and please accept my apology for the delay in contacting you with a decision. The paper has been reviewed by two subject experts and a statistician; their comments are included below and can also be accessed here: [LINK]

As you will see, the reviewers found the study interesting, but they raised a few questions and requested some clarification of some of the methodological details. After discussing the paper with the editorial team and an academic editor with relevant expertise, I'm pleased to invite you to revise the paper in response to the reviewers' comments. We plan to send the revised paper to some or all of the original reviewers, and we cannot provide any guarantees at this stage regarding publication.

In view of the upcoming holidays, we ask that you submit your revision by Monday, January 6th. However, if this deadline is not feasible, please contact me by email, and we can discuss a suitable alternative. Please also feel free to contact me directly with any questions (hvanepps@plos.org). 

Kind regards, 

Heather 

Heather Van Epps, PhD 

Executive Editor

PLOS Medicine

hvanepps@plos.org

Comments from the editors/academic editor:

1. We felt that the discussion could be expanded to further contextualize the findings relative to real-world clinical scenarios; for example, the scenario of GPs providing no clinical explanation to patients with knee OA (ie, radiographic explanation only) is not well-documented, and it would be useful cite any previous studies that address this (current citations focus more on prescription trends for imaging). 

2. The current data availability statement does not comply with the PLOS Data Availability Policy (see http://journals.plos.org/plosmedicine/s/data-availability), which requires that all data underlying the study's findings be provided in a repository or as Supporting Information. For data residing with a third party, authors are required to provide instructions with contact information (web or email address) for obtaining the data. Please note that a study author cannot be the contact person for the data. PLOS journals do not allow statements supported by "data not shown" or "unpublished results." For such statements, authors must provide supporting data or cite public sources that include it.

3. Abstract: please include details of the planned statistical analyses (primary and secondary comparisons).

4. Please include the full recruitment dates (Month, Day, Year) in the Abstract, with beginning and ending dates. 

5. We take an all-or-none approach to inclusion of secondary outcomes in the abstract. If you choose to include secondary outcomes, they must all be included (methods and findings) to avoid selective reporting. Given the large number of secondary outcomes, we prefer that the abstract be limited to the primary outcome (per CONSORT). 

6. Methods: please check the URL provided for the SAP, as the link does not appear to be correct/functional. 

7. Please consider moving Figure 2 to the supplement, as we were not convinced that this graphic adds to the numerical reporting of the outcomes (although it would be very useful for oral presentations). 

8. When you submit your revision, please include the original protocol (uploaded as a supplemental file) and the SAP (if a separate document). It was an oversight that these documents were not requested prior to peer review. 

Comments from the reviewers: 

Reviewer #1: Statistical review

This paper reports a three-arm RCT comparing hypothetical consultation strategies for individuals with knee osteoarthritis. I found the study design to be an interesting approach to investigate beliefs. The statistical methods are suitable and are mostly well-reported. I had only some minor comments, provided below:

1. Abstract: it would be useful to clarify that the first outcome is also a 11 point NRS - at the moment it is not clear that this is the case and it could be that it's a binary outcome. Perhaps add 'both' before 'measured on' on line 51.

2. Abstract: I would recommend providing the mean NRS per group as well as the mean difference + confidence interval. I would also provide this for the exercise/physical activity outcome.

3. Abstract: given the number of secondary outcomes in the trial registration page, I would recommend either not reporting results of secondary outcomes in the abstract, or briefly mentioning them all e.g. 'no significant differences were found for other outcomes including…'.

4. Table 3: I would recommend reporting p-values to 2 significant figures or <0.001 (I think this is the PLOS guidance).

James Wason

Reviewer #2: 

This is a really interesting study on the potential effect of radiographic diagnosis on patients' beliefs about the management of their knee OA. I commend the authors for undertaking a well-designed and well-reported study examining the differences between clinical and radiographic diagnoses (without and with showing x-rays) using a hypothetical scenario. I have only a few minor queries about this work. 

1. I wondered if the authors had any information or indication on why people failed to complete the survey, as this was quite a large proportion? What implications could this have had on the findings of the trial?

2. Previous research in electronic health records databases has shown that lots of patients with knee pain don't get a diagnosis of knee OA at their first consultation for their knee pain/knee problem. In the trial, patients who had previously consulted for knee pain were understandably excluded from this trial; however, their experience may also be important to consider, and I feel this should be acknowledged.

3. Can the authors comment on the impact of negative social media and negative messages that still exist in society, such as 'wear and tear', and the role they may have played on the study participants' beliefs and the expectations that a joint replacement will be needed, in general, but also with reference to the those who received a radiographic diagnosis? 

Reviewer #3: 

This is a very interesting paper. In my opinion this is a well-conducted and thorough study. I have only minor points. 

1. Abstract: Clearly written abstract, this highlights the study's main results. It probably won't fit to add the nuance of the third group. 

2. Methods: As I understand correctly you used a company to recruit people. Do they get paid for filling in a survey? 

3. The link to the SAP is not working. 

4. In the text I do not read if the GP is the same in each video, I guess so, but it is not mentioned. 

5. Results: Figure 1 is clear. I was confused about the text saying 617 participants were enrolled and 523 were excluded. May be rephrase a bit.

---

* Please upload any figures associated with your paper as individual TIF or EPS files with 300dpi resolution at resubmission; please read our figure guidelines for more information on our requirements: http://journals.plos.org/plosmedicine/s/figures. While revising your submission, please upload your figure files to the PACE digital diagnostic tool, https://pacev2.apexcovantage.com/. PACE helps ensure that figures meet PLOS requirements. To use PACE, you must first register as a user. Then, login and navigate to the UPLOAD tab, where you will find detailed instructions on how to use the tool. If you encounter any issues or have any questions when using PACE, please email us at PLOSMedicine@plos.org.

* Please ensure that the study is reported according to the CONSORT guideline and include the completed CONSORT checklist (https://www.equator-network.org/reporting-guidelines/consort/) as Supporting Information. When completing the checklist, please use section and paragraph numbers, rather than page numbers. Please add the following statement, or similar, to the Methods: "This study is reported as per CONSORT guideline (S1 Checklist)."

FIGURES AND TABLES

SUPPLEMENTARY MATERIAL

REFERENCES

RCTs

* PLOS Medicine requires that all trials be prospectively registered in one of registries recognized by WHO. Please ensure that study registration details are included in the Methods section.

* Please structure the Methods section using the following sub-headings: Study design and participants, Randomization and masking, Procedures, Outcomes, Statistical analysis.

* Please ensure that all prespecified outcomes (primary, secondary, and exploratory) are listed in the Methods/Outcomes section and indicate whether there are outcomes that are not presented in the current report.

* Please specify the dates (Month Day, Year) during which study enrollment and follow up occurred.

* Please include absolute numbers wherever you report percentages; eg, n/N (%)

* Please present the safety data for the study including numbers of specific events and whether or not adverse events are thought to be related to treatment. AEs should be reported in the abstract, per CONSORT and CONSORT-Harms.

* Please ensure that all components of CONSORT are present in the manuscript, including how randomization was performed, allocation concealment, blinding of intervention, definition of lost to follow-up, power statement. As above, when completing the checklist, please use section and paragraph numbers, rather than page numbers.

* Please report your abstract according to CONSORT for abstracts, following the PLOS Medicine abstract structure (Background, Methods and Findings, Conclusions) https://www.equator-network.org/reporting-guidelines/consort-abstracts/

* In keeping with our commitment to Open Science, please include the study protocol document and analysis plan (including any amendments) as Supporting Information to be published with the manuscript if accepted.

* Please note that PLOS Medicine requires prospective, public registration of a data sharing plan (as part of mandatory clinical trials registration) for all clinical trials that began enrollment on or after January 1, 2019, in accordance with ICMJE requirements.

---

## [Editor Report · Decision Letter 2]

15 Jan 2025

Dear Dr. Lawford,

Thank you very much for re-submitting your revised manuscript "Effects of x-ray based diagnosis and explanation of knee osteoarthritis on patient beliefs about osteoarthritis management: A randomised clinical trial." (PMEDICINE-D-24-03202R2).

I have discussed the paper with my colleagues and the academic editor, and we were happy with the revised submission and your responses to the reviewers’ comments. As such, we did not feel that a second round of peer review was necessary. I am pleased to say that provided the remaining editorial and production issues are dealt with we are planning to accept the paper for publication in the journal.

[LINK]

In revising the manuscript for further consideration, please ensure you address the specific points made by the editors. In your rebuttal letter you should indicate your response to the reviewers' and editors' comments and the changes you have made in the manuscript. Please submit a clean version of the paper as the main article file. A version with changes marked must also be uploaded as a marked up manuscript file.

We expect to receive your revised manuscript within 1 week (Jan 22nd). Please email me directly (hvanepps@plos.org) if this deadline is not feasible and we can discuss a suitable alternative. Please also reach out with any questions or concerns.

Kind regards,

Heather

Heather Van Epps, PhD

Executive Editor 

PLOS Medicine

plosmedicine.org

Requests from Editors:

1. Abstract, line 45 (and author summary). Please consider replacing ‘considered’ with ‘were presented with’ (ie, “Participants were presented with a hypothetical scenario…”). 

2. Abstract line 53-54. If you wish to include the secondary comparison (clinical explanation [no x-rays] versus radiographic explanation [not showing x-ray images]) in the methodology paragraph, you should report the corresponding outcomes. Alternatively, you can remove the secondary comparison from the abstract (per CONSORT). 

3. Please add a sentence at the end of the Abstract/Methods and findings section indicating the main limitation(s) of the trial. 

4. Please review your text for claims of novelty or primacy (e.g. 'for the first time') and either remove this language or include “to our knowledge”—for example in the author summary and Discussion, line 390. In addition, please check that any use of statistical terms (such as trend or significant) are supported by the data, and if not please remove them.

5. Please remove subheadings from the Discussion section, which should be presented as a continuous narrative. 

6. Please include URLs for relevant references (eg, refs 5, 6, 18, etc). 

7. Please ensure that the link to the Dryad Digital Repository is live when you submit the final revision.

8. Please modify the CONSORT checklist to use section and paragraph numbers, rather than page numbers. 

9. Please upload the supplemental material as a separate file to the main manuscript. 

10. In accordance with ICMJE requirements, PLOS Medicine requires prospective, public registration of a data sharing plan (as part of mandatory clinical trials registration) for all clinical trials that began enrollment on or after January 1, 2019.

---

## [Editor Report · Decision Letter 3]

20 Jan 2025

Dear Dr Lawford, 

On behalf of my colleagues and the Academic Editor, Christelle Nguyen, I am pleased to inform you that we have agreed to publish your manuscript "Effects of x-ray based diagnosis and explanation of knee osteoarthritis on patient beliefs about osteoarthritis management: A randomised clinical trial." (PMEDICINE-D-24-03202R3) in PLOS Medicine.

PRESS

Kind regards,

Heather

Heather Van Epps, PhD 

Executive Editor 

PLOS Medicine